# Legal Retrieval for Public Defenders

## Abstract

AI tools are suggested as solutions to assist public agencies with heavy workloads. In public defense—where a constitutional right to counsel meets the complexities of law, overwhelming caseloads, and constrained resources—practitioners face especially taxing conditions. Yet, there is little evidence of how AI could meaningfully support defenders' day-to-day work. In partnership with the *anonymized Office of the Public Defender*, we develop the *anonymized BriefBank*, a retrieval tool which surfaces relevant appellate briefs to streamline legal research and writing. We show that existing retrieval benchmarks fail to transfer to real public defense research, however adding domain knowledge improves retrieval quality. This includes query expansion with legal reasoning, domain-specific data and curated synthetic examples. To facilitate further research, we release a taxonomy of realistic defender search queries and a manually annotated evaluation dataset for public defense retrieval. This benchmark is highly correlated with a proprietary retrieval dataset annotated by experienced public defenders. Our work improves on the status quo of realistic legal retrieval benchmarking and illustrates one approach to applying AI in a real-world public interest setting.

## 1 Introduction

In the United States, individuals facing criminal charges have a right to counsel. This right is guaranteed by the Sixth Amendment and reaffirmed in Gideon v. Wainwright (1963). For those unable to afford private counsel, representation is provided by public defenders. In practice however, defenders often face severe resource constraints and overwhelming caseloads (Pace et al., 2023), while having to navigate the complexities of today's legal system. Combined, these can undermine promises of fair and equal legal representation for clients relying on public defense.

Advances in natural language processing, particularly in foundation models, have raised hopes that AI tools could assist public defenders by streamlining time-consuming tasks, including for example legal research or drafting of briefs (Bommasani et al., 2022; Mahari et al., 2023; Cheong et al., 2025). However, despite rapid advances in model capabilities and legal benchmark performance (Guha et al., 2023; Dominguez-Olmedo et al., 2025), there remain very few examples of concrete real-world implementation and evaluation of AI within public defender offices.

This gap limits our understanding of what types of AI use cases are feasible, safe, and can genuinely empower defenders in day-to-day legal practice. Public defense is a high-stakes setting where errors can directly affect the outcome of cases and clients. Hence, carefully balancing trade offs between accuracy, reliability and risks of failure become essential design constraints in such applications. Consequently, any AI assistance for public defenders must prioritize verifiable and trustworthy outputs.

In this work, we partner with the *anonymized Office of the Public Defender (anonymized OPD)* to identify, develop, and evaluate such an AI use case. Defenders often specialize in specific legal areas, like felonies or misdemeanors. If handed a case outside their area of specialization, they often consult colleagues to obtain past briefs within the office handling similar cases. Such past briefs, especially those from the appellate section written by experienced defenders, offer overviews, reusable legal arguments, applicable precedent and overall guidance for how to navigate similar legal circumstances.

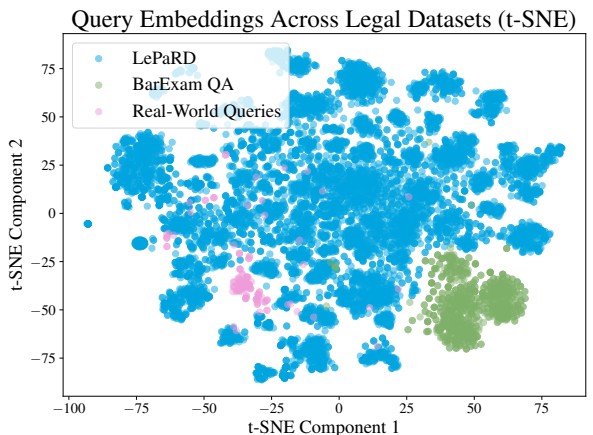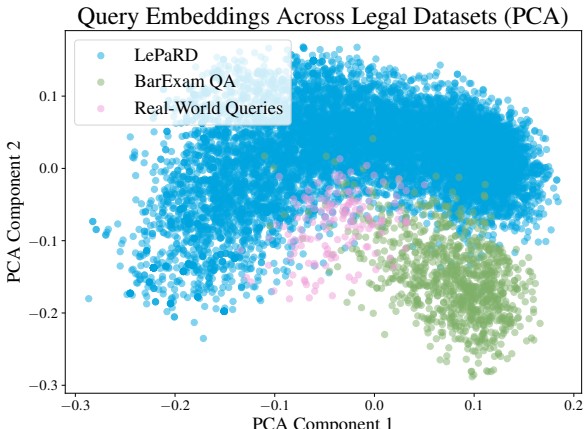

Figure 1: Visualizations of defender queries (pink dots) and other queries in two legal search datasets: LePaRD (Mahari et al., 2024) and BarExam-QA (Zheng et al., 2025). We compute low-dimensional projections with t-SNE (van der Maaten & Hinton, 2008) and PCA (Pedregosa et al., 2011). We observe that queries from different datasets are separable in the embedding spaces.

Inspired by such current office practices, we developed the *anonymized BriefBank*. The BriefBank leverages foundation model-based embeddings for retrieval, along other components, to search over all appellate briefs within the office and surfaces relevant ones. By providing access to relevant briefs, defenders can reuse legal arguments, precedent and other applicable information. Thus, the BriefBank is designed to streamline brief drafting, but also maintains more consistency within the office: defenders may be more likely to leverage already-identified winning strategies and follow best practices.

During multiple evaluation rounds, experienced defenders submitted realistic search queries and assessed the quality of the BriefBank. The submitted queries are highly diverse: They include broad topical searches, specific legal arguments, pinpoint citations, statutory definitions and doctrinal status checks. These queries reflect the heterogeneous and practice-driven information needs of public defenders.

In contrast, existing legal retrieval datasets are often constructed somewhat artificially, e.g., by reusing bar-exam questions as queries (Zheng et al., 2025), finding similar cases given a query case (Kim et al., 2023), or by heuristically linking preceding context to quoted legal passages (Hou et al., 2025; Mahari et al., 2023). In Appendix Table 3, we show that no existing U.S. legal retrieval datasets contain both realistic queries and manually verified target paragraphs. While existing datasets are valuable for controlled benchmarking and measuring progress on legal retrieval, they poorly approximate more realistic contexts. Figure 1 visualizes query embeddings from BarExam-QA (Zheng et al., 2025), LePaRD (Mahari et al., 2024), and public defender queries, which are separable in embedding space. In Section 5, we will show that training on such datasets decreases performance on public defense retrieval.

On the other hand, we improve recall by (1) using more recent and larger embedding models (Lee et al., 2024; Zhang et al., 2025b) and (2) leveraging domain knowledge. To introduce domain knowledge, we generate curated synthetic training data using a fine-tuned defender query generation model, and filter that data with a fine-tuned legal reranker (similar to Dai et al., 2023). We further expand queries in this dataset using the IRAC framework[1] developed for legal analysis. See Section 5 for full results.

Our work is a first step toward practical AI for public defense, inspired by current office practices. Ott et al. (2022) argue that future benchmarks should emphasize real-world utility. In our work, we offer a step towards more realistic legal NLP benchmarking, enabled by collaboratively developing the Briefbank in partnership with the *anonymized OPD*. To stimulate further research on legal retrieval for public defenders, we release a manually annotated dataset, fine-tuned models and replication code.[2]

---

[1] Issue, Rule, Application, and Conclusion.

[2] anonymized replication package: https://anonymous.4open.science/r/anonymized-public-defender-retrieval-782E

Combined, these artifacts could help develop similar briefbanks for other defender offices or pro-bono clinics, but also measure progress in more realistic legal retrieval settings. More broadly, this collaboration illustrates how partnerships with public institutions allow situating NLP research in real-world applications. To summarize, we make the following contributions:

- We introduce the public defense retrieval task: retrieving relevant passages from existing appellate defense briefs. We construct an accompanying evaluation dataset comprised of 170 queries and 543 human-annotated relevant paragraphs from publicly available documents (Section 3).
- Evaluation of eight pre-trained retrieval models on public defense search. While larger models perform better, the best model only achieves 37.08% recall@5, demonstrating substantial room for improvement on this task (Table 2).
- A taxonomy of defender search queries, discussing shortcomings of current models and informing future work on public defense search (Section 4).
- Evidence of a mismatch between existing legal retrieval benchmarks and real-world use cases. We find that fine-tuning retrieval models on existing legal retrieval benchmarks degrades performance, while legal domain adaptation, fine-tuning on carefully curated synthetic data and query expansion strategies improve performance (Section 5).

## 2 Public Defense Retrieval

Cheong et al. (2025) group public defense work into five pillars, of which two seem especially suitable to AI assistance: evidence investigation and legal research and writing. For legal research and writing, they report that AI would be most useful to generate surveys of information, provide starting points, draft documents and narrow down case searches.

This perspective is reinforced in our collaboration with the *anonymized OPD*. A well-written brief functions as a structured survey of a legal issue. The office encourages defenders to use such briefs as starting points if they have to handle a case outside their specialization. Thus, access to such briefs streamlines legal drafting through reuse of existing materials. Lastly, these briefs contain relevant legal precedent, and thus indirectly narrow down case search.

Apart from searching for briefs, we also explored other AI use cases to empower defenders. These include (1) directly answering legal queries using generative AI with a closed-source RAG tool (NotebookLM), and (2) searching through federal and state case law. However, both failed to deliver sufficient levels of accuracy, verifiability, and transparency, hence we define public defense retrieval as searching over past briefs, materials defenders already use and trust. Building accurate search over briefs also serves as a lens into the collective knowledge acquired within an office over time, and makes that knowledge accessible.

### 2.1 Retrieval vs. Generation

Past work pointed out potential usefulness of generative AI for legal work (Bommasani et al., 2022; Mahari et al., 2023), which has been supported by promising legal evaluation of LLMs, such as GPT-4 passing the bar exam (Katz et al., 2024). Schwarcz & Choi (2023) find that LLMs help law students draft legal documents faster. Given such evidence, the office explored the potential of LLMs to assist public defenders in their day-to-day work. Experienced defenders submitted 100 queries to NotebookLM, a Retrieval-Augmented Generation (RAG) application. Although NotebookLM had access to a small set of relevant briefs, 66% of the NotebookLM generations contained issues. The three main failure modes consisted of:

1. Hallucinations (of e.g., citations) or incorrect references to source materials.
2. Failure of the model to address nuanced legal contexts.
3. Incomplete or verbose outputs, or generating unrelated information altogether.

Strikingly, Cheong et al. (2025) report that 85% of public defenders they interviewed currently doubt AI can reliably verify research output, describing the same reasons (hallucinations, failure to handle nuanced legal context and incomplete output) the OPD identified in their explorative evaluation. Given the high stakes of public defense, where hallucinated citations and misstated precedents can affect the outcome of cases,

Table 1: Legal search queries and associated paragraphs

| Search Query | Relevant Paragraph |
|---|---|
| Difference between reasonable suspicion and probable cause | *[Anonymized case citation]* Although reasonable suspicion is a less demanding standard than probable cause, "[n]either 'inarticulate hunches' nor an arresting officer's subjective good faith can justify infringement of a citizen's constitutionally guaranteed rights. |
| does the fruit of the poisonous tree doctrine apply in the fifth amendment context, too? | See *[Anonymized law]* ("The fruit-of-the-poisonous-tree doctrine denies the prosecution the use of derivative evidence obtained as a result of a Fourth or Fifth Amendment violation. "). |
| Is consent to search valid if the motor vehicle stop was illegal? | Because the police unlawfully prolonged the detention and sought consent to search without reasonable suspicion of criminal activity, in violation of the Fourth Amendment and Article I, paragraph 7 of the state Constitution, all of the evidence must be suppressed. *[Anonymized state law]*; Wong Sun v. United States, 371 U.S. 471, 484 (1963). As "warrantless stops and searches are presumptively invalid, the State bears the burden of establishing that any such stop or search is justified by one of the 'well-delineated exceptions' to the warrant requirement. " *[Anonymized state law]*. The law is clear that when a car is stopped due to a purported motor-vehicle violation, "[a]uthority for the seizure . . . ends when tasks tied to the traffic infraction are – or reasonably should have been – completed. |

such failure rates are unacceptable in practice. Beyond accuracy, using commercial generative models raises confidentiality risks: case details submitted to proprietary APIs may fall outside attorney-client privilege and be subject to mandatory disclosure (Cheong et al., 2024; Paine & Travisano, 2025). Generative models also provide limited transparency about sources, making it difficult for attorneys to verify output accuracy.

## 2.2 Brief vs. Case Law Retrieval

We also considered retrieval over all state and federal case law, which would directly return relevant precedent. However, this risks surfacing opinions that are no longer good law. As prior work shows, approximately 7.8% of lower-court decisions are later reversed on appeal (Edwards, 2019). Because defenders cannot rely on non-binding or outdated precedent, case law search alone without information about whether a case is still good law seems too unreliable for practical use cases. While commercial systems include such information, automatically detecting overturned case law remains an active and unresolved research problem (Zhang et al., 2025a). In the BriefBank, we address this by including internal documents and public directives, both containing, among others, overturned case alerts and up-to-date best practices for frequently occurring legal issues.

## 2.3 Task Definition

Formally, we define the public defense retrieval task as follows: Given a user query $q$, which may be related to a citation, rule reference, doctrinal question, or natural-language description of a legal concept, the goal is to retrieve the most relevant paragraphs $p_i$ from a corpus of prior briefs, other internal documents and public directives. Each query can have multiple relevant paragraphs.

# 3 The *anonymized BriefBank* and Corresponding Retrieval Dataset

In this section, we describe the BriefBank in more detail, and the associated PD dataset construction process.

## 3.1 BriefBank Overview

The BriefBank is a retrieval system that enables defenders to search across the office's internal corpus of appellate briefs, directives, and guidelines to locate relevant passages for new cases. The primary goal of the system is to make the collective institutional knowledge of the office, including arguments, citations, and legal reasoning, accessible within seconds. This corpus consists of 2896 briefs spanning the last 25 years, 168 internal documents and 351 public directives. We automatically split these into 140K unique paragraphs using LLM-based semantic segmentation (Smith & Troynikov, 2024).

When a user submits a query, the BriefBank retrieves relevant paragraphs and presents them alongside contextual metadata (such as brief title and filing date). Each retrieved passage is accompanied by an LLM-generated summary of the legal issue and case facts to help users decide which returned briefs warrant

closer inspection. Importantly, generation is only used to summarize existing content and help defenders quickly decide whether a brief might be relevant, not to create new arguments or citations, preserving factual reliability.

The system prioritizes recent briefs, as newer materials are more useful. Search results are thus sorted by recency. Future versions may incorporate instruction-following rerankers that also account for other metadata, for example whether a brief contributed to a successful outcome for a client.

Over three evaluation rounds during development, experienced public defenders have submitted 194 queries to the BriefBank. After each submitted query, they could provide detailed feedback: binary search result annotation (relevant or irrelevant) for up to five retrieved paragraphs, and additional freeform textual feedback. They annotated 85.6% of all returned paragraphs, from which 66% were annotated as being relevant for public defense work. Moreover, they provided textual feedback for 55.2% of the queries, which further contextualize the annotations and search results.

## 3.2 The Public Defense Dataset

Due to confidentiality constraints, the internal OPD dataset cannot be released. To support reproducible research, we construct the **Public Defense Dataset** (PD dataset), a resource that mirrors the structure and characteristics of the proprietary dataset, using the same search queries, but target paragraphs from publicly available documents. We show dataset examples and annotated paragraphs in Table 1.[3]

The PD dataset contains:

- 170 authentic public defender queries, collected from multiple evaluation rounds with senior public defenders in *anonymized state* (discarding all queries containing personally identifiable information).
- 96,032 unique paragraphs segmented using LLM-based semantic chunking, the same process applied to OPD's internal corpus. They are drawn from a corpus of 856 publicly available documents, including appellate briefs and state or Attorney General directives.
- 543 human-annotated relevant paragraphs manually annotated and verified by the author team. All annotators have ample background in legal NLP, and half of the annotators are currently in or have completed law school.

**Dataset construction.** To build this publicly releasable corpus, we scraped state and AG guidelines, and briefs from cases argued at the appellate level between 2023 and 2025. In total, this results in 856 documents: 351 directives and 505 briefs. We convert all pdfs to text using olmOCR (Poznanski et al., 2025). Afterwards, we apply LLM-based semantic segmentation (Smith & Troynikov, 2024) to split the documents into 96,032 unique paragraphs.

We then obtained candidate search results for all queries obtained during BriefBank evaluation rounds from this corpus (discarding all queries containing personally identifiable information). For each query in the OPD set, we retrieve the following candidates: We collect the 100 most similar paragraphs to the query using the NV-Embed model (Lee et al., 2024) and the 10 most similar paragraphs based on a keyword search. If we have annotated gold paragraphs from an evaluation round, we also retrieve the 10 most similar paragraphs for each gold paragraph. In total, each query yields 110–160 paragraph candidates: 100 from the NV-Embed, 10 from BM25, and 0–50 from the most similar paragraphs retrieved from annotated results. We narrow down the set of potential candidates using LLMs. First, we use GPT-4 as a judge to filter paragraphs which are potentially irrelevant. We then rerank all remaining paragraphs using a Qwen3-8B reranker (Zhang et al., 2025b) fine-tuned on the proprietary OPD dataset.

Next, we manually review up to seven highest scoring paragraphs (if all paragraphs were discarded by the reranker or GPT-4o, we discard the query), and decide whether they are a good search result. For this annotation, we take into account the query, our own legal expertise, and the already annotated gold paragraphs from the BriefBank evaluation rounds and additional feedback collected from that evaluation round. The annotation was performed by the author team, all with backgrounds in legal NLP. Additionally, half of the annotators are in or have completed law school. To compute inter-annotator agreement, an

---

[3]The full dataset and replication code to rerun experiments conducted in this work will be released upon publication.

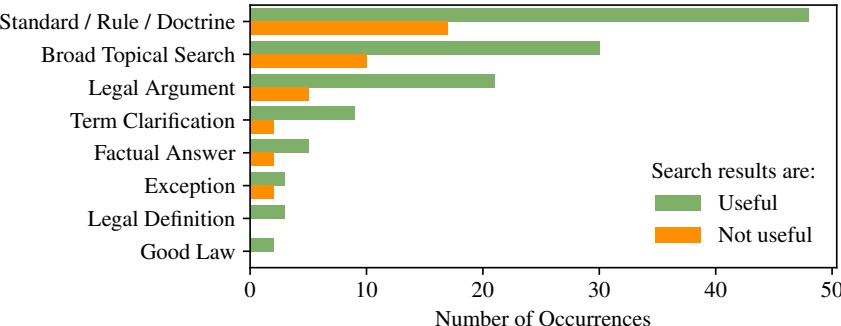

Figure 2: Anonymized BriefBank's performance on public defender queries categorized by **search intent**. Most queries fall into standard/ rule/ doctrine, broad topical search, and legal argument.

independent annotator with law school experience also annotated 100 query/paragraph pairs, using the same annotation guidelines. Annotations between the author team and that annotator result in a Cohen's Kappa of 0.36, indicating fair agreement. Most of these disagreements (66%) can be explained by the independent annotator being more lenient, and annotating slightly relevant paragraphs too. The author team was more conservative and often rejected slightly relevant paragraphs instead.

Lastly, we anonymize party-related personally identifiable information in the dataset. We use the 31B Gemma 4 model (Google DeepMind, 2026) to anonymize all paragraphs, using the prompt shown in Appendix A. This results in 147,044 anonymized entities (1.5 entities per paragraph). To evaluate the anonymization procedure, we manually verified 50 paragraphs, 25 where no information was anonymized and 25 paragraphs were at least one entity was anonymized: we did not find any personally identifiable information (PII) in paragraphs where the model did not anonymize any information. For the anonymized entities, we find that 67% of anonymizations are correct, while 28% contain over-anonymization: judge names or organization names insufficient to identify parties. In 5% of cases the model anonymizes non-relevant legal terms (e.g., case index numbers in a Table of Citations).

Performance on the two datasets (the proprietary OPD and the publicly released PD Dataset) is highly correlated: Across eight zero-shot experiments, recall@5 results in a spearman R of 0.79 (p = 0.02) and across 20 fine-tuning experiments of retrieval models, the spearman R is 0.84 (p=9.1e-6), which we will discuss in more detail in Section 5. We will further show that anonymization does barely affects retrieval performance: the spearman R between the anonymized and non-anonymized version is 1.0 in zero-shot settings (perfect correlation), and 0.99 (p=2.0e-16) for fine-tuning experiments. In Appendix Table 4, we further present dataset statistics from both PD and the proprietary OPD dataset and show that quantifiable metrics in the two datasets seem comparable. Combined, these findings make us confident that the PD dataset captures relevant signals about public defender search, and may also be used as a more realistic evaluation set to benchmark NLP and AI methods against.

## 4  Taxonomy of Defender Queries

To better understand the PD dataset, we manually annotated all queries for search intent (what a defender was searching for) and algorithmic search strategies (keyword-based, embedding-based, and agentic). We then construct a taxonomy, which includes information about how often queries have been successfully answered by the BriefBank. These can point to common failure cases in realistic legal retrieval. While commercial companies have such data at scale, little to no publicly available information on real-world legal queries can be found. We describe the annotation process in more detail in Appendix Table 5.

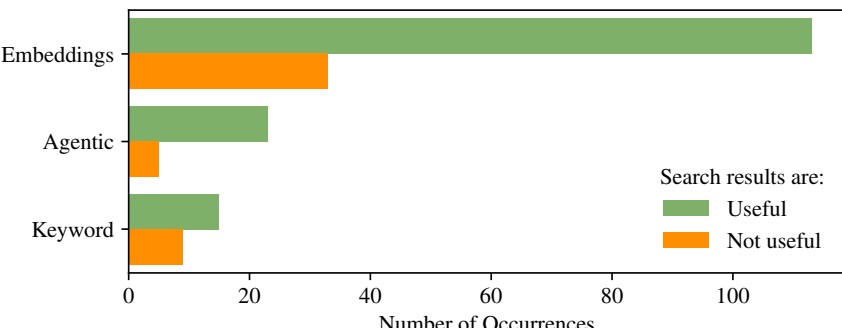

Figure 3: BriefBank's performance on public defender queries categorized by **required search strategy**. Most queries only require embedding-based retrieval. Queries that require keyword-based retrieval is the most challenging for the current system.

## 4.1 Search Intent

We find substantial variation in public defense search queries, ranging from queries asking for specific cases or rules by title, e.g., *"anonymized statute"*, to more natural language queries, e.g., *"Find me briefs about inevitable discovery?"* to questions about legality, e.g., *"are tinted windows legal in anonymized state"* to complex queries requiring multihop retrieval and reasoning, such as *"has Counterman v. Colorado been addressed in a published anonymized state opinion?"*. Overall, we identify eight broad categories of search objectives and plot the distributions across them in Figure 2. The majority of queries fall into either (1) legal standards, rules or doctrines, (2) search of legal arguments or briefs about certain topics, and (3) a few less frequently used categories.

**Standards, Rules, and Doctrines.** Queries that ask for the legal standard or rules, often for a specific situation, for example *"standard for ordering passenger out of a car"* or *"anonymized statute"*. These are the most frequent and often expressed through natural language or keyword-based queries. They closely reflect the day-to-day needs of defenders when conducting legal research and writing. These queries often include keyword-based queries which embedding-based retrieval approaches struggle with.

**Topical or Argument-Oriented Searches.** Queries that broadly look for briefs or passages about a certain topic, such as *"find briefs about community caretaking"*, or specifically for legal arguments, for example *"What are arguments against consent searches during illegal car stops?"*. Having relevant briefs about certain topics can give an overview of the legal landscape and applicable legal arguments. Failure cases include a lack of ability in current models to distinguish nuanced legal contexts: Consider the query "reverse 404b". Reverse 404(b) is when a defense lawyer introduces evidence of another person's past acts to exonerate their client, whereas the standard 404(b) rule is typically used by prosecutors to introduce a defendant's past acts to prove guilt. Embedding models for this query always return results only about the standard 404(b) rule.

Other, less frequent categories include term clarifications (e.g., *"difference between reasonable suspicion and probable cause"*), definitions and exceptions (e.g., *"booking exception to Miranda"*), factual or procedural questions (e.g., *"when was anonymized act amended?"*) and questions about good law (e.g., *"Is anonymzed case citation still good law?"*). Failure cases here can be summarized as the model not sufficiently understanding the query, or the relevant information not being in the BriefBank.

## 4.2 Information Sources and Search Strategies.

We also annotate the required search strategy for each query. We group search strategies into embedding-based, keyword-based and agentic search (Figure 3):

Table 2: Evaluation metrics (zeroshot)

| Model | OPD Dataset (%) | | PD Dataset | |
|---|---|---|---|---|
| | Recall 1 | Recall 5 | Recall 1 | Recall 5 |
| all-mpnet-base-v2 | 6.79 | 19.72 | 7.44 | 19.30 |
| e5-base-v2 | 11.11 | 27.44 | 6.27 | 25.23 |
| e5-large-v2 | 11.34 | 29.61 | 8.67 | 27.4 |
| Qwen3-Embedding-0.6B | 8.69 | 30.93 | 9.28 | 29.35 |
| Qwen3-Embedding-4B | 10.33 | 36.84 | 11.13 | 34.19 |
| e5-mistral-7b-instruct | 14.48 | 41.97 | 11.11 | 32.61 |
| NV-Embed-v2 | **15.12** | **51.85** | 11.48 | 31.27 |
| Qwen3-Embedding-8B | 13.16 | 40.19 | **13.37** | **37.08** |
| fine-tuned e5-large-v2 | 10.72 | 33.71 | 10.40 | 36.26 |

Evaluation metrics (%) for various retrieval models with Recall1 and Recall5 on the proprietary OPD and the PD dataset. Rows sorted by model size, above the dashed line models with less than 1B parameters, below with more than 1B parameters. Last row shows results of a fine-tuned e5-large model on synthetic data with query expansion, approaching the performance of the 25x bigger Qwen3-Embeddings-8B model. See section 5 for more details.

- Embedding-based retrieval computes dense vector representations for queries and paragraphs, and the paragraphs closest to a query are returned as results.
- Keyword-based retrieval, such as BM25 (Lù, 2024) represent queries and documents as sparse vectors, capturing some statistic about word frequency such as TF-IDF weights. Search results are again paragraphs with the highest similarity to a query.
- Agentic search strategies (Deng et al., 2023; OpenAI, 2025) refer to methods that broadly involve an agent conducting retrieval, with actions such as query expansion (Zheng et al., 2025), multihop retrieval, and reasoning over multiple retrieval steps.

Embedding-based retrieval captures most queries, but keyword searches remain common due to defenders' familiarity with boolean-style legal research systems. Notably, keyword-style queries have the highest rate of unhelpful results (38%), suggesting that models optimized for natural language retrieval still underperform when users employ traditional legal search syntax.

One illustrative example for agentic queries is *"has Counterman v. Colorado been addressed in a published anonymized state opinion"*. To answer this query, one would first need to retrieve all published opinions referring to Counterman v. Colorado, requiring a search index over all state case law, and associated metadata about publishing status. Then, the agent would need to read all these opinions and decide whether they sufficiently address the main arguments in Counterman v. Colorado, and finally return the answer. Optimally, the agent would provide excerpts from the state opinions addressing the case, along with links to the full opinions. We believe this points to exciting avenues for future work in AI-powered public defense research.

## 5 Empirical Evaluation of Retrieval Models and Rerankers

We evaluate two components of the anonymized BriefBank pipeline: (1) retrieval models, which encode both queries and paragraphs into embeddings and return a set of candidate results based on semantic similarity, and (2) rerankers, which re-score the top-$k$ retrieved passages using more expressive cross-encoder LLMs.

**Experimental Setup** All experiments are conducted on both the internal OPD dataset obtained during the BriefBank evaluation rounds, and the PD dataset described in Section 3. Each paragraph in the datasets is treated as an independent retrieval unit. Given the size of the datasets, we consider the whole dataset as a test set only and report results on that test set. For all fine-tuning experiments, we report the mean result of five runs with different seeds, and confidence intervals in the Appendix.

### 5.1 Retrieval Experiments

We evaluate retrieval performance of eight pre-trained retrieval models: all-mpnet-v2 (Reimers & Gurevych, 2019), e5-base-v2 and e5-large-v2 (Wang et al., 2022), Qwen3-Embedding-0.6B, -4B and-8B (Zhang et al., 2025b), e5-mistral-7b-instruct (Wang et al., 2023) and NV-Embed-v2 (Lee et al., 2024). We use Recall@k (with $k = 1, 5$) as the metric. This metric is informative for practitioner-facing search systems where defenders typically inspect only the top few results.

**Zero-shot retrieval.** We first evaluate zero-shot performance of eight pre-trained models (Table 2). We observe that larger models (above 4B parameters) perform better than smaller models (below 1B parameters) on the OPD dataset and the PD dataset. The results also confirm that PD is a good approximate of the internal OPD dataset. Performance the two datasets is correlated, with a spearmanR of 0.78 (p=0.02). If we discard results from the NV-embed model, we obtain a spearman R of 0.89 (p=0.007). The NV-Embed model is potentially confounding, as the first version of the BriefBank simply returned the top five paragraphs found by the NV-Embed model. By construction, that model has a recall of 100% on all these examples, which in turn inflates performance numbers for this model on the OPD dataset.

**Fine-tuning on existing legal retrieval benchmarks.** Next, we fine-tuned four smaller models (all-mpnet-v2, e5-base-v2, e5-large-v2, Qwen3-Embedding-0.6B) on two existing legal retrieval benchmarks: BarExam-QA and LePaRD. All reported results are the mean of five independent fine-tuning runs with different seeds. Fine-tuning on these datasets leads to a decrease in Recall@5. In Figure 4, we show averaged performance gains (or losses). We show exact results for all models in Appendix Table 7. To illustrate, the blue bars indicate the effect of further fine-tuning models on the BarExam-QA dataset. If evaluated on the same BarExam-QA dev set, performance increases by 1.7 points in Recall5, compared to the zero-shot performance of the same models. Training on BarExam-QA also slightly increases performance on LePaRD. However, performance on both the proprietary OPD and the released PD dataset decreases.

**Fine-tuning on naive synthetic dataset.** Since fine-tuning on existing benchmarks does not lead to performance gain, we experiment with synthetic datasets. We first construct and evaluate a "naive" synthetic dataset generated by a Llama3-70B model (Grattafiori et al., 2024). To construct the naive synthetic dataset, we generate a corresponding search query for each paragraph in our corpus. To prevent data leakage in this synthetic dataset, we remove all paragraphs which are annotated retrieval targets. We include four (query, paragraph) pairs from the OPD dataset as few-shot examples to guide generation. The model prompt is shown in Appendix Figure 6. Fine-tuning on this dataset increases performance on BarExam-QA and LePARD, indicating that there is some signal about legal similarity in that synthetic dataset. However, training on this dataset substantially decreases performance on the two public defense test sets.

**Fine-tuning on optimized synthetic dataset.** Next, we construct an optimized synthetic dataset by fine-tuning the generation model on domain specific data. We fine-tune a Llama3-70B model using annotated (query, paragraph) pairs that are obtained from BriefBank evaluation rounds. Input to the model is an annotated paragraph, output is the query for which the paragraph was retrieved. We use the same system prompt (shown in Appendix Figure 6). During this fine-tuning process, the model learns how to generate more realistic queries. Next, we filter all (query, paragraph) pairs with a fine-tuned Qwen3-reranker trained on the OPD dataset, and discard all generated examples below a certain threshold. After further inspection of the dataset, we find that paragraphs containing facts, tables of contents, or other procedural content rarely appear in the annotated search results. Thus, we also filter out such paragraphs using a zero-shot Llama3-70B model. Eventually, fine-tuning on the resulting synthetic dataset improves performance on public defense datasets.

**Query Expansion.** Following (Zheng et al., 2025), we also experimented with query expansion strategies. Using Llama-70B, we expand each query by first applying the IRAC framework (issue, rule, application, conclusion), a well known method for legal analysis: spot the issue, identify the relevant legal rule, apply the rule to the issue, draw the conclusion. After IRAC, we then derive an expanded search query. Input to the model is the concatenation of the original query, the IRAC analysis and the expanded query. We show

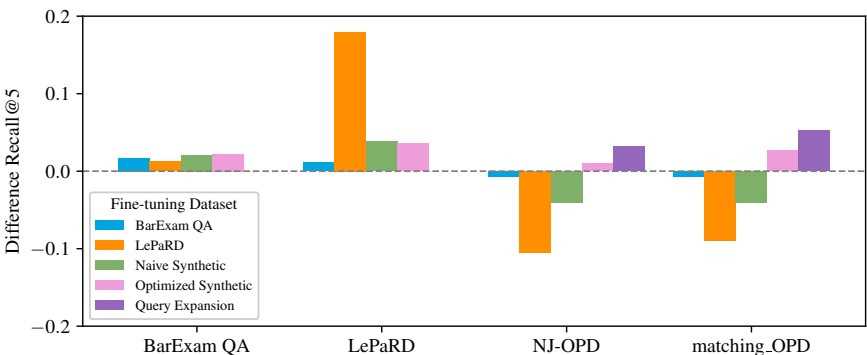

Figure 4: **Average gains or loss in recall @ 5 for four retrieval models.** Colored bars indicate model performance after fine-tuning on different retrieval datasets. On the x-axis, we plot resulting model performance on four retrieval datasets (BarExam QA, LePaRD, OPD and PD). Changes in recall are relative to a zero-shot baseline of the same model. If fine-tuned on BarExam QA, LePaRD or naive synthetic data, performance on public defense datasets decreases. if fine-tuned on carefully tuned synthetic data, performance on public defense datasets increases. Full results are shown in Appendix Table 7.

the system prompt and one augmented query example in Appendix Table A. We expand all public defense queries in the test set, and all curated synthetic queries. Next, we fine-tune models on the expanded, curated synthetic set. The e5-large-v2 approaches the performance of the larger Qwen3-Embedding-8B model in this regime of fine-tuning with expanded queries. In the zero-shot setting, we find mixed results where query expansion increases recall for smaller models, but recall decreases for larger model.

**Overall comparison.** Leveraging domain knowledge, we can improve retrieval quality. We show that query expansion eliciting IRAC traces and an optimized synthetic dataset both lead to performance increases. However, training on existing legal benchmarks and naively generated synthetic data decreases quality. We speculate this is caused by a domain shift: these datasets are simply too different from public defense retrieval. We alluded to this phenomenon in Figure 1 and provide further evidence by showing queries from all datasets in Appendix Table 13. We show exact results (instead of average performance) for all models in Appendix Table 7.

**Robustness of the PD Dataset** Results between the proprietary OPD dataset and the released PD dataset are highly correlated. In zero-shot settings, the resulting spearman R is 0.78 (p=0.02) for recall@5, for fine-tuning experiments, the spearman R consists of 0.82 (p=9.1e-6) across 20 fine-tuning experiments (four models and five different training datasets). We also report and compare to retrieval results on a non-anonymized version of the PD dataset, and report results in Appendix Table Ÿ11. The resulting spearman R of the anonymized and non-anonymized version is 1.0 for the zero-shot setting (perfect correlation), the correlation of the 20 fine-tuning experiments results in a spearman R of 0.99 (p=2.0e-16). Zero-shot correlations between the non-anonymized version and the proprietary OPD dataset are the same, the correlation of fine-tuning experiments decreases slightly to 0.82 (from 0.88). We believe these results together confirm the validity of the benchmark, although it has been annotated by the author team and undergone anonymization.

Since OPD documents span 25 years, and our PD dataset only contains briefs from 2023–2025, we also verify that retrieval performance is robust across different time periods of the OPD dataset. We report spearman R between the PD dataset and OPD subsets stratified by year in Appendix Table 6. Correlations between the released PD dataset and the OPD dataset are not driven by old OPD retrieval targets, but also generalize to more recent retrieval targets.

For all experiments, we fine-tune models five times with different seeds and we report the mean results of these five runs. In Appendix Table 8, we show the 95% confidence intervals, and note that in almost all

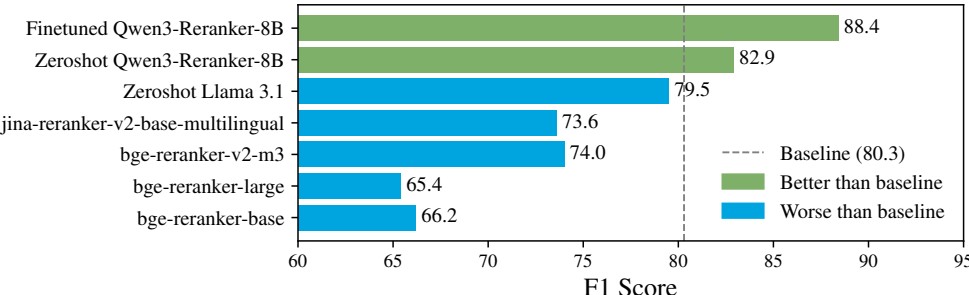

Figure 5: **F1 scores of different reranker models.** Δ is the difference from the majority-baseline F1 (most annotated paragraphs are relevant, so a majority baseline achieves a high F1 score). Most off-the-shelf rerankers perform worse than the majority baseline on detecting good vs. bad search results.

experiments, differences to zero-shot results of the same base model are significant. The confidence intervals of the fine-tuning experiments are moderate, with 0.70% points for recall@5 for the OPD dataset, and 0.83% points for the PD dataset, and thus both comparable to the confidence intervals of the BarExam QA dataset (0.68% points).

Given the practitioner-facing nature of the PD dataset, we report recall@5 as the official metric for the PD dataset (the BriefBank by default also returns five search results). Recall@5 is highly correlated with other standard information retrieval metrics, including Normalized Discounted cumulative gain (NDGC@5), Mean Reciprocal Rank (MRR@10), and Mean Average Precision (MAP@100). The lowest of these correlations with the reported recall@5 results is in the PD dataset and the MRR@10 metric, where the spearman R is 0.97 (p=2.6e-12). We show detailed results for all metrics in Appendix Table 9.

**Domain Adaptation.** We additionally experiment with legal domain adaptation, another method to add domain knowledge to models. We further pre-train a ModernBERT-large checkpoint on 30B tokens of US case law opinions using the masked language modeling objective. This domain-adapted checkpoint, if further fine-tuned on the optimized synthetic dataset, achieves a Recall@5 of 23.4, compared to a vanilla ModernBERT-large checkpoint fine-tuned on the same dataset, which achieves 18.7 instead (-4.7). We show exact results in Appendix Table 10.

## 5.2 Reranker Experiments

We evaluate six open-source rerankers on the OPD dataset. We additionally fine-tune the most performant model on the OPD dataset, showing that this further improves performance. We use an 80-20 split to separate queries into a training and test set (stratified by queries, to make sure that no training queries appear in the test set). We then calculate the F1 score for all models.

Table 12 shows the reranking results of all models. We compare all results to a simple majority baseline. The baseline assumes a simple heuristic which treats each search result as a good result. We plot the F1 score of this majority baseline as a horizontal dashed line (at 80.02% F1). We observe that most existing rerankers underperform the simple majority baseline, with the only exception being the recently released Qwen-3-8B reranker (Zhang et al., 2025b), which slightly surpasses this baseline. In contrast, fine-tuning on domain-specific data leads to significant improvements. This suggests that legal search for public defenders is primarily a data problem. Detailed metrics (precision, recall, accuracy, and F1) for all models are provided in Appendix Table 12.

# 6 Discussion

## 6.1 Difference between Existing Benchmarks and Public Defense Retrieval

We find a distribution mismatch between existing legal retrieval datasets and our public defender search dataset, where training on legal benchmarks (Zheng et al., 2025; Mahari et al., 2024) actually lowers model performance. Similarly, Gu et al. (2025) note that outdated medical domain benchmarks are inadequate to assess current AI, and Ott et al. (2022) emphasize that most existing benchmarks lack real-world utility. Especially for public defense work, there seem to be no suitable benchmarks to advance AI and NLP methods, but also most legal retrieval datasets do not contain manually verified retrieval targets, and none contain real-world queries (Appendix Table 3. The academic field of NLP and AI for public good and access to justice is growing (Karamolegkou et al., 2025; Mahari et al., 2023; 2024), yet limited by data availability.

To make progress on this front, we provide several starting points to stimulate further research on public defense retrieval. We release the PD dataset containing realistic queries drafted by experienced public defenders, and manually verified corresponding paragraphs relevant to those queries. Performance on the PD dataset is correlated with a proprietary dataset created by public defenders. Next, we construct a taxonomy about search objectives of defenders and what search strategies can be employed to answer these queries. Similar to datasets like WildChat Zhao et al. (2024), which make real-world chatGPT conversations accessible for research, we hope our queries and taxonomy can inform future work in more realistic legal retrieval settings.

## 6.2 Collaborations between Academia and Legal Institutions

This work illustrates a collaboration between academic researchers and a public agency. We believe such partnerships can be mutually beneficial: Public agencies get to be involved in reflecting on their existing workflows, and identifying suitable AI use cases, and gain exposure to the opportunities, trade-offs, risks and barriers of of AI applications.

For researchers, such collaborations allow work on AI tasks and use cases that are more closely aligned with real institutional needs and constraints, and are often absent in existing datasets and AI benchmarks, especially in the legal domain (See Table 3). Through frequent meetings and discussions, knowledge transfer benefits both sides: agencies clarify their AI needs, and researchers gain insight into tacit constraints that can inform better AI methods.

We believe such collaborations represent a promising research direction to work on more realistic AI applications with practical utility (Ott et al., 2022). As NLP and AI techniques mature and increasingly promise real-world impact, progress is often limited by the lack of realistic datasets and evaluation settings. We view our collaboration with the *anonymized OPD*, and accompanying data artifacts, as one step toward addressing this gap.

## 6.3 Future Work To Improve Public Defense Retrieval

Zheng et al. (2025) find substantial gains leveraging query expansion and legal reasoning to retrieve relevant statutes for bar exam questions. We confirm this for public defense retrieval, and believe there's value in exploring such efforts in more detail. Moreover, we believe agentic search (Deng et al., 2023), which combines multiple retrieval steps into a single agentic workflow. During this process, the agent can explore, among others, query expansion, legal reasoning, multihop retrieval, and reranking, until a suitable search result is retrieved.

Second, the current frontier embedding models, i.e., e5-mistral (Wang et al., 2023), Qwen3-Embedding (Zhang et al., 2025b) and NV-Embed-V2 (Lee et al., 2024) already perform substantially better than smaller, older models built on top of BERT or RoBERTa models, as as shown in Table 2. We believe further such advances, and especially advances in dedicated legal retrieval models, may further increase performance on public defense retrieval.

Synthetic data likely can be leveraged to further improve model performance in legal retrieval. However, we note two challenges: The first is that synthetic data must be carefully curated, as we have shown in Section 5. In Section 2, we described how the OPD has experimented with a RAG tool, but ultimately rejected the idea because models often were imprecise in addressing specific legal questions. It seems that current models seem to lack the ability to handle nuanced legal contexts (see also Cheong et al., 2025), which might affect synthetic data. Further research will be needed to make progress on this front.

## 6.4 Related Work

Our work contributes to a growing academic field on how to use legal NLP and AI in collaboration with public agencies. Related work includes AI assistance for automatically detecting and redacting racial covenants (Surani et al., 2025), automatically clearing records at scale (Code for America, 2020) or AI assistance for eviction defense (Stanford Law School Legal Design Lab, 2025). Similar to our work, these projects also identified a suitable use case for AI assistance, and in collaboration with public agencies developed specialized methods to accomplish the goal.

Moreover, we release a dataset for legal retrieval, and contribute to the academic literature on legal retrieval in the United States (Kim et al., 2023; Mahari et al., 2024; Hou et al., 2025; Zheng et al., 2025). In contrast to these works, we focus on retrieval for public defenders, where the goal is to retrieve relevant paragraphs from legal briefs. Following the recommendations made in Ott et al. (2022), we have put an emphasis on real-world utility while creating this benchmark.

We do acknowledge the potential of NLP and AI methods for other types of public defense work. Cheong et al. (2025) outline a research agenda for how AI can assist public defenders, and foremost identify use cases around making sense of large volumes of data in evidence investigation. Similar to our work, public defenders in (Cheong et al., 2025) report similar challenges of using AI for legal research and writing, however this can change rapidly as technology advances.

## 7 Conclusion

In this paper, we discuss legal retrieval for public defenders. In collaboration with the *anonymized OPD*, we identify retrieval over internal briefs as a suitable use case to assist public defense work, and developed the *anonymized BriefBank*. This tool allowed us to gather realistic public defense queries, from which we manually construct and release the PD retrieval dataset.

Query expansion, carefully curated synthetic datasets and legal domain adaptation increase retrieval performance, while in-domain fine-tuning increases reranking accuracy. However, training on existing academic legal retrieval datasets lowers performance, indicating a distribution shift between these benchmarks and the more realistic PD retrieval task. Our results suggest that progress in legal retrieval for public defense may be constrained less by model scale, but by domain mismatch and lack of available datasets.

## 8 Broader Impact Statement

This work introduces public defense retrieval as an NLP task and releases an associated benchmark. We design a retrieval-based BriefBank to surface relevant appellate briefs to support defenders' day-to-day legal research, while taking into account high stakes and low error tolerance of public defense work. The tool is deliberately designed to not be a AI decision-making system, but AI assistance mimicking existing office practices. The tool serves as an AI interface to the office's collective institutional knowledge and mirrors existing word-of-mouth practices.

The BriefBank avoids using generative AI for substantive legal tasks. Exploratory evaluation revealed unacceptable failure rates in RAG-based approaches (hallucinations, failure to handle nuanced legal contexts), consistent with qualitative findings defenders report in Cheong et al. (2025). LLM-generated summaries are included in the tool only to help defenders triage results. We implement multiple additional safeguards to encourage responsible usage: (1) users are instructed to not rely on these summaries. We include one full

paragraph discussing generative AI risks and the summaries in the user instructions of the BriefBank. (2) The summaries are not visible by default, but have to be expanded. (3) If expanded, a user first sees a disclaimer (in bold and large font-size) that the following summary is AI-generated and might be incomplete, factually incorrect or contain hallucinations. (4) We added CSS safeguards such that these summaries cannot be copy-pasted to discourage summaries from being incorporated in new documents.

The released PD dataset enables future research in more realistic legal retrieval settings. Performance on the public dataset is strongly correlated with a proprietary defender-annotated dataset, suggesting that it provides a valid signal for benchmarking. We hope it also serves as a starting point for similar briefbank deployments at other defender offices. However, we acknowledge that the benchmark stems from a single office in one U.S. state. Generalization to other jurisdictions or office contexts should thus be verified. Moreover, the benchmark might be biased (1) towards retrieval targets specific to *anonymized state*, (2) specific annotation idiosyncrasies of the annotating defenders and the author team, and (3) we relied on automatic filtering using an ensemble of GPT-4o and a fine-tuned Qwen3-8B reranker to further reduce the number of paragraphs we manually reviewed. These limitations need to be considered in future usage.

Finally, any deployment of AI tools in public defense must carefully consider confidentiality. Using proprietary APIs risks falling outside attorney-client privilege and may be subject to mandatory disclosure (Cheong et al., 2024; Paine & Travisano, 2025). The BriefBank was deployed as a closed-universe system using open models following our discussion with the *anonymized OPD*. We recommend other projects developing access to justice AI tools to take into account constraints around sensitive data, privilege and confidentiality, and to adopt similar safeguards.

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

# A   Appendix

Table 3: Comparison of queries and paragraphs in legal retrieval datasets

| Dataset | Domain | Real queries | Manually verified targets |
|---|---|---|---|
| LePaRD (Mahari et al., 2024) | US case law | ✗ | ✗ |
| CLERC (Hou et al., 2025) | US case law | ✗ | ✗ |
| Law Search as Prediction (Dadgostari et al., 2021) | US case law | ✗ | ✗ |
| AirBench (Chen et al., 2025) | Pile of Law | ✗ | ✗ |
| BVA citation prediction (Huang et al., 2021) | Board of Veterans' Appeals | ✗ | ✗ |
| Contract Summarization (Manor & Li, 2019) | Contracts | ✗ | ✗ |
| COLIEE (Kim et al., 2023) | US case law / statutes | ✗ | ✓ |
| BarExam-QA (Zheng et al., 2025) | US case law / bar exams | ✗ | ✓ |
| PD Dataset (ours) | Appellate briefs | ✓ | ✓ |

Comparison of queries and paragraphs in legal retrieval datasets. This list includes all retrieval datasets used in Legal-Bench (Guha et al., 2023), all U.S. legal retrieval datasets in MTE-Bench (Muennighoff et al., 2023) and all U.S. datasets in MLEB (Butler et al., 2025).

Table 4: Dataset statistics, comparing the released dataset to the proprietary OPD dataset.

| Statistic | Proprietary Dataset | Released Dataset |
|---|---|---|
| Number of queries | 194 | 170 |
| Average gold paragraphs per query | 2.9 | 3.2 |
| Average query length (words) | 8.9 | 9.3 |
| Average paragraph length (words) | 133.9 | 155.0 |
| Type-Token Ratio (TTR) | 0.15 | 0.13 |

Table 5: Examples of Queries and Annotations for the purposes of deriving a taxonomy.

| Query | Objective | Search Strategy |
|---|---|---|
| Standard for ordering passenger out of a car | standard | embeddings |
| *anonymized statute* | rule | keyword |
| inevitable discovery | doctrine | keywords / embeddings |
| find briefs about community caretaking | topical search | keywords / embeddings |
| What are arguments against consent searches during illegal car stops? | legal argument | embeddings |
| Difference between reasonable suspicion and probable cause | term clarification | embeddings |
| when was *anonymized act* amended to enumerate the offenses subject to its provisions? | Factual Answer | agentic |
| has Counterman v. Colorado been addressed in a published anonymized state opinion? | Factual Answer | agentic |
| booking exception to miranda | exception | embeddings |
| what is the definition of probable cause? | definition | embeddings |
| Is *anonymized case* still good law? | good law | embeddings / agentic |

Examples of Queries and Annotations for the purposes of deriving a taxonomy. We manually annotate all queries in our dataset. While annotating, we consider the following information: The query itself, relevant search results annotated by experienced public defenders for a query which help us understand what they were searching for, and additional freeform textual feedback. Examples in this table were used as annotation guidelines).

---

**Generating Synthetic Queries**

You are a helpful legal assistant. You are given a paragraph from a legal brief. Your task is to come up with a question / query for which the paragraph would be the top search result. Make sure that the query is not too specific.
Some examples are shown below.
*Examples omitted due to data confidentiality reason*
In case the snippet is not detailed enough or doesn't contain neither facts nor legal reasoning, just reply with None, else return the generated query, nothing else."""

Figure 6: System prompt to generate synthetic data. User prompts are all paragraphs in the briefs dataset.

---

**Anonymizing Paragraphs**

You are a PII extraction assistant. You are given a passage from a legal document (a court brief or opinion).
Your task is to identify all personally identifiable information (PII) related to parties in a brief excerpt, and return them as a JSON array of unique strings.
PII to extract contains: - Names of individuals (parties, judges, attorneys, witnesses)
- Names of specific companies or organizations when they are a party or can identify a party
- Street addresses, cities, ZIP codes when they identify a party
- Docket numbers and case numbers when they identify parties
- Dates of birth, Social Security numbers, phone numbers, email addresses when they identify parties
Do NOT include:
- Generic legal terms, statutory references without jurisdiction identifiers, or procedural terminology
- Legal Citations
- Names that appear within a case citation (e.g., [anonymized] should not be extracted) - Statutory citation abbreviations such as "[anonymized]"
when used as part of a statute reference (e.g., "[anonymized]") — these are legal citation shorthand, not identifying references
Return only a JSON array of strings — the unique text spans of each identified PII item — with no explanation, no commentary, and no markdown
formatting.

Figure 7: System prompt for anonymizing paragraphs. User prompts are all paragraphs in the dataset.

---

**Annotation Guidelines**

Each data point consists of a query submitted by the anonymized OPD and a retrieved paragraph from state directives or briefs. Your task is to annotate whether the paragraph is useful in public defense work given query. Label each pair as either useful or not useful.
A paragraph is useful if it would plausibly help a public defender who entered that query—for example, if it is topically relevant, cites relevant statutes or precedent, or provides information related to the issue implied by the query. Mentally group each result into no, low, moderate, or high relevance, and mark it as useful if it is moderate or high. If the paragraph is clearly unrelated or unhelpful, label it not useful. For keyword-based queries (e.g., names like "anonymized name"), mark a result as useful if the keyword appears in the paragraph and the content is topically appropriate. If the intent of a query is unclear (e.g., "merger,"), flag the example as unclear intent in a comment.
For each data point, consider the query and retrieved paragraph carefully. You also often see gold paragraphs, examples previously annotated by experienced OPD defenders, and any accompanying textual feedback. These should be used as reference points to calibrate your understanding of what constitutes a useful result and to maintain consistency across annotations.

Figure 8: Annotation guidelines for annotating the PD Dataset.

---

**System Prompt for IRAC Query Expansion**

Given a legal search query, you should:
1. Infer the key legal issue(s) raised by the query.
2. State the applicable legal rule(s) in general doctrinal terms.
3. Optionally provide a brief legal analysis (reasoning) if it helps clarify the issue and rule.
4. Construct an augmented search query that incorporates the original query plus useful legal reasoning signals (issues, rules, key concepts, doctrinal terms). This augmented query may be in any style (keywords, IRAC-style summary, or a well-structured legal question), as long as it is helpful for retrieving relevant cases and statutes.
Important constraints:
- Do NOT invent or guess specific statute numbers, code sections, or guideline provisions unless they already appear in the original query.
- Do NOT introduce new facts, parties, or jurisdictions that are not clearly implied by the query.
- You MAY generalize to doctrinal labels and concepts (e.g., "merger of offenses", "double jeopardy", "premises liability").
- The augmented query MUST include the original query content in some form (verbatim or lightly edited) plus additional legally-relevant language.
- Be concise but not minimal: prefer adding a few high-value issues/rules/terms over verbose filler.
Output format:
issue: "concise statement of the main legal issue"
rule: "concise statement of the applicable legal rule or doctrine",
analysis: "optional brief reasoning, 1–3 sentences, may be empty if not helpful",
augmented query: "the final expanded query string used for retrieval"
Remember:
- The 'augmented query' can be in natural language or semi-structured (e.g. IRAC-style, dense prose, or keyword-enriched), but it must be optimized to help a legal search engine retrieve the most relevant authorities.
- Do NOT mention statutes or guideline provisions by number if they are not already present in the original query.

Figure 9: System prompt for IRAC query expansion. User prompts are all queries in the dataset.

Table 6: Spearman R correlations of zero-shot retrieval performance (Recall@5, 8 models) between OPD subsets stratified by document release date and the PD dataset

| OPD Period | Queries | Retrieval Targets | Spearman R (p-value) |
|---|---|---|---|
| All years (overall) | 194 | 563 | 0.79 (0.02) |
| 1998–2019 | 114 | 357 | 1.00 (<0.001) |
| 2020–2025 | 70 | 137 | 0.88 (0.004) |
| 2023–2025 | 46 | 66 | 0.61(0.108) |

The OPD corpus spans 25 years and its annotated retrieval targets are predominantly from older briefs; the PD dataset covers documents from 2023–2025. Strong correlations across all time periods indicate that retrieval performance on annotated paragraphs of older OPD documents generalizes to newer ones. The marginally significant Spearman R of 0.61 between the PD dataset and the OPD subset is likely driven by the small dataset size of the OPD subset. For certain documents, we do not have the year information, hence we discard these in this stratified analysis.

Table 7: Recall@5 results obtained via training models on different legal retrieval datasets.

| Train Dataset | Model | BarExam QA | LePaRD | Internal OPD | PD Dataset |
|---|---|---|---|---|---|
| BarExam QA | all-mpnet-base-v2 | 4.52 (+2.90) | 15.75 (+1.44) | 21.80 (+2.07) | 19.67 (+0.37) |
| | e5-base-v2 | 3.23 (-1.61) | 16.82 (+0.47) | 28.06 (+0.62) | 25.46 (+0.23) |
| | e5-large-v2 | 3.23 (+1.61) | 18.64 (+1.11) | 32.03 (+2.43) | 28.11 (+0.71) |
| | Qwen3-0.6B | **11.94** (+3.87) | 20.51 (+1.41) | 22.66 (-8.26) | 25.37 (-3.98) |
| LePaRD | all-mpnet-base-v2 | 3.39 (+1.77) | 32.03 (+17.71) | 18.17 (-1.56) | 17.35 (-1.96) |
| | e5-base-v2 | 4.84 (+0.00) | 28.82 (+12.47) | 9.59 (-17.85) | 7.46 (-17.77) |
| | e5-large-v2 | 5.97 (+4.35) | 36.52 (+18.99) | 20.01 (-9.59) | 19.41 (-7.99) |
| | Qwen3-0.6B | 7.10 (-0.97) | **41.70** (+22.60) | 18.08 (-12.84) | 21.04 (-8.32) |
| Naive Synthetic | all-mpnet-base-v2 | 5.97 (+4.35) | 19.81 (+5.49) | 23.39 (+3.67) | 22.42 (+3.12) |
| | e5-base-v2 | 3.39 (-1.45) | 18.62 (+2.28) | 20.37 (-7.07) | 17.57 (-7.66) |
| | e5-large-v2 | 8.71 (+7.10) | 21.14 (+3.60) | 25.35 (-4.25) | 23.76 (-3.64) |
| | Qwen3-0.6B | 6.61 (-1.45) | 22.90 (+3.80) | 22.46 (-8.47) | 21.45 (-7.91) |
| Optimized Synthetic | all-mpnet-base-v2 | 7.90 (+6.29) | 19.68 (+5.37) | 27.28 (+7.56) | 24.53 (+5.23) |
| | e5-base-v2 | 3.23 (-1.61) | 18.23 (+1.88) | 27.99 (+0.54) | 24.42 (-0.80) |
| | e5-large-v2 | 7.26 (+5.65) | 21.28 (+3.74) | 29.84 (+0.24) | 31.09 (+3.68) |
| | Qwen3-0.6B | 6.77 (-1.29) | 22.31 (+3.21) | 26.54 (-4.38) | 31.99 (+2.64) |
| Query Expansion | all-mpnet-base-v2 | n/a | n/a | 31.89 (+12.17) | 26.48 (+7.18) |
| | e5-base-v2 | n/a | n/a | 31.65 (+4.21) | 27.82 (+2.6) |
| | e5-large-v2 | n/a | n/a | **33.71** (+4.10) | **36.26** (+8.86) |
| | Qwen3-0.6B | n/a | n/a | 27.03 (-3.90) | 35.32 (+5.97) |

Recall@5 results obtained via training models on different legal retrieval datasets. In brackets: difference to zero-shot version of the same model. All models trained with `sentence-transformers` (Reimers & Gurevych, 2019), using a learning rate of 2e-5, a batch size of 128, and the CachedMultipleNegativesRankingLoss (Gao et al., 2021). Best performance by benchmark in bold (Qwen3-0.6B trained on BarExam QA for BarExam, Qwen3-0.6B trained on LePaRD for LePaRD, e5-large-v2 trained on query expansions for both public defender search datasets.)

Table 8: Recall@5 results with confidence intervals

| Train Dataset | Model | BarExam QA | LePaRD | Internal OPD | PD Dataset |
|---|---|---|---|---|---|
| BarExam QA | all-mpnet-base-v2 | 4.52 (±0.9)** | 15.75 (±0.04)** | 21.80 (±0.25)** | 19.67 (±0.32)** |
| | e5-base-v2 | 3.23 (±0.0)** | 16.82 (±0.03)** | 28.06 (±0.32)** | 25.46 (±0.54)** |
| | e5-large-v2 | 3.23 (±0.0)** | 18.64 (±0.03)** | 32.03 (±0.64)** | 28.11 (±0.21)** |
| | Qwen3-0.6B | **11.94** (±0.84)** | 20.51 (±0.07)** | 22.66 (±0.60)** | 25.37 (±0.73)** |
| LePaRD | all-mpnet-base-v2 | 3.39 (±0.45)** | 32.03 (±0.02)** | 18.17 (±0.33)** | 17.35 (±0.46)** |
| | e5-base-v2 | 4.84 (±0.0)** | 28.82 (±0.02)** | 9.59 (±0.61)** | 7.46 (±0.47)** |
| | e5-large-v2 | 5.97 (±1.82)** | 36.52 (±0.08)** | 20.01 (±0.67)** | 19.41 (±1.29)** |
| | Qwen3-0.6B | 7.10 (±1.49) | **41.70** (±0.17)** | 18.08 (±1.29)** | 21.04 (±1.27)** |
| Naive Synthetic | all-mpnet-base-v2 | 5.97 (±0.55)** | 19.81 (±0.03)** | 23.39 (±0.15)** | 22.42 (±0.33)** |
| | e5-base-v2 | 3.39 (±0.45)** | 18.62 (±0.05)** | 20.37 (±0.48)** | 17.57 (±0.45)** |
| | e5-large-v2 | 8.71 (±1.10)** | 21.14 (±0.12)** | 25.35 (±0.98)** | 23.76 (±2.25)** |
| | Qwen3-0.6B | 6.61 (±1.31)** | 22.90 (±0.11)** | 22.46 (±1.63)** | 21.45 (±1.67)** |
| Optimized Synthetic | all-mpnet-base-v2 | 7.90 (±0.45)** | 19.68 (±0.04)** | 27.28 (±0.41)** | 24.53 (±0.63)** |
| | e5-base-v2 | 3.23 (±0.0)** | 18.23 (±0.08)** | 27.99 (±0.36)** | 24.42 (±0.99) |
| | e5-large-v2 | 7.26 (±0.71)** | 21.28 (±0.08)** | 29.84 (±1.37) | 31.09 (±0.55)** |
| | Qwen3-0.6B | 6.77 (±0.90)** | 22.31 (±0.05)** | 26.54 (±0.62)** | 31.99 (±0.49)** |
| Query Expansion | all-mpnet-base-v2 | n/a | n/a | 31.89 (±0.46)** | 26.48 (±0.74)** |
| | e5-base-v2 | n/a | n/a | 31.65 (±0.71)** | 27.82 (±0.95)** |
| | e5-large-v2 | n/a | n/a | **33.71** (±1.09)** | **36.26** (±0.53)** |
| | Qwen3-0.6B | n/a | n/a | 27.03 (±1.06) ** | 35.32 (±1.85)** |

Recall@5 results obtained via training models on different legal retrieval datasets. In brackets: 95% confidence intervals across 5 independent fine-tuning runs with different seeds. All models trained with `sentence-transformers` (Reimers & Gurevych, 2019), using a learning rate of 2e-5, a batch size of 128, and the CachedMultipleNegativesRankingLoss (Gao et al., 2021). Best performance by benchmark in bold (Qwen3-0.6B trained on BarExam QA for BarExam, Qwen3-0.6B trained on LePaRD for LePaRD, e5-large-v2 trained on query expansions for both public defender search datasets.)

Table 9: Different IR evaluation metrics for public defender datasets

| Train Dataset | Model | Internal OPD | | | | PD Dataset | | | |
|---|---|---|---|---|---|---|---|---|---|
| | | R@5 | NDCG@5 | MRR@10 | MAP@100 | R@5 | NDCG@5 | MRR@10 | MAP@100 |
| BarExam QA | all-mpnet-base-v2 | 21.80 | 21.62 | 36.46 | 21.00 | 19.76 | 19.34 | 31.16 | 17.96 |
| | e5-base-v2 | 28.06 | 29.30 | 49.30 | 27.44 | 25.41 | 23.25 | 35.48 | 21.40 |
| | e5-large-v2 | 32.03 | 33.22 | 51.29 | 32.09 | 28.16 | 26.56 | 40.38 | 24.58 |
| | Qwen3-0.6B | 22.66 | 21.37 | 35.56 | 21.53 | 25.51 | 23.83 | 36.58 | 22.43 |
| LePaRD | all-mpnet-base-v2 | 18.17 | 18.80 | 33.96 | 18.24 | 17.28 | 16.21 | 26.73 | 15.52 |
| | e5-base-v2 | 9.59 | 9.52 | 18.37 | 8.98 | 7.54 | 7.37 | 13.22 | 7.03 |
| | e5-large-v2 | 20.01 | 19.36 | 34.30 | 17.95 | 19.58 | 19.06 | 31.53 | 17.21 |
| | Qwen3-0.6B | 18.08 | 17.45 | 30.94 | 16.46 | 21.41 | 19.60 | 29.43 | 18.60 |
| Naive Synthetic | all-mpnet-base-v2 | 23.39 | 23.60 | 39.14 | 23.62 | 22.42 | 20.32 | 29.32 | 19.35 |
| | e5-base-v2 | 20.37 | 20.78 | 34.20 | 19.82 | 17.53 | 16.11 | 25.58 | 16.13 |
| | e5-large-v2 | 25.35 | 24.98 | 39.87 | 24.22 | 23.78 | 22.05 | 34.75 | 21.52 |
| | Qwen3-0.6B | 22.46 | 21.58 | 36.02 | 21.47 | 21.50 | 20.61 | 33.06 | 20.22 |
| Optimized Synthetic | all-mpnet-base-v2 | 27.28 | 27.71 | 46.07 | 27.74 | 24.63 | 22.88 | 34.89 | 22.81 |
| | e5-base-v2 | 27.99 | 28.71 | 46.46 | 27.49 | 24.44 | 23.53 | 36.64 | 22.93 |
| | e5-large-v2 | 29.84 | 29.85 | 46.70 | 30.35 | 30.80 | 28.37 | 42.26 | 27.92 |
| | Qwen3-0.6B | 26.54 | 26.67 | 42.89 | 26.71 | 31.88 | 29.25 | 42.45 | 28.53 |
| Query Expansion | all-mpnet-base-v2 | 31.89 | 34.72 | 55.58 | 34.52 | 26.54 | 24.99 | 38.15 | 25.51 |
| | e5-base-v2 | 31.65 | 31.86 | 48.58 | 30.85 | 27.94 | 26.08 | 38.85 | 25.56 |
| | e5-large-v2 | 33.71 | 33.90 | 51.29 | 34.45 | 36.0 | 32.28 | 46.23 | 30.48 |
| | Qwen3-0.6B | 27.03 | 27.36 | 44.21 | 27.78 | 35.10 | 32.11 | 45.62 | 31.19 |
| **Spearman R** | | – | 0.989 | 0.983 | 0.979 | – | 0.992 | 0.968 | 0.974 |

Table 10: Domain adaptation results

| Train Dataset | Model | BarExam QA | LePaRD | NJ-OPD | PD Dataset |
|---|---|---|---|---|---|
| BarExam QA | ModernBERT-large | 1.61 (±0.00) | 7.30 (±0.06) | 1.02 (±0.18) | 0.26 (±0.00) |
| | Legal-ModernBERT | **2.42** (±**0.00**) | **8.77** (±**0.04**) | 1.18 (±0.00) | **1.78** (±**0.23**) |
| LePaRD | ModernBERT-large | 3.55 (±0.90) | 32.16 (±0.12) | 6.85 (±0.40) | 5.25 (±0.60) |
| | Legal-ModernBERT | 3.23 (±0.00) | **32.41** (±**0.12**) | **9.46** (±**0.90**) | **9.89** (±**0.17**) |
| Naive Synthetic | ModernBERT-large | 3.71 (±0.55) | 17.30 (±0.11) | 10.88 (±0.27) | 11.75 (±1.11) |
| | Legal-ModernBERT | 4.03 (±0.00) | **17.80** (±**0.05**) | **12.48** (±**0.51**) | **16.0** (±**0.83**) |
| Optimized Synthetic | ModernBERT-large | 2.42 (±0.00) | 17.01 (±0.08) | 20.34 (±0.74) | 18.74 (±0.53) |
| | Legal-ModernBERT | **4.19** (±**0.45**) | **17.42** (±**0.05**) | 20.82 (±1.11) | **23.39** (±**0.83**) |

Recall@5 with 95% confidence intervals (mean ± CI over five seeds). Bold indicates statistically significant differences bewteen Legal-ModernBERT and ModernBERT-large, based on non-overlapping confidence intervals. All models trained with `sentence-transformers` using a learning rate of 2e-5, batch size 128, and CachedMultipleNegativesRankingLoss.

Table 11: Comparison Retrieval Performance (recall@5) on anonymized and non-anonymized PD Dataset

| Experiment | Anonymized | Non-anonymized | $\delta$ (difference) |
|---|---|---|---|
| all-mpnet-base-v2 | 19.30 | 19.55 | − 0.25 |
| e5-base-v2 | 25.23 | 25.06 | + 0.17 |
| e5-large-v2 | 27.40 | 28.25 | − 0.85 |
| Qwen3-Embedding-0.6B | 29.35 | 29.83 | −0.48 |
| Qwen3-Embedding-4B | 34.19 | 34.10 | +0.09 |
| e5-mistral-7b-instruct | 32.61 | 33.63 | −1.02 |
| NV-Embed-v2 | 31.27 | 31.93 | −0.66 |
| Qwen3-Embedding-8B | 37.08 | 37.37 | −0.29 |

Spearman R of 1.0 (perfect correlation) for retrieval performance between anonymized and non-anonymized version of the PD Dataset (zero-shot settings). Spearman R across 20 fine-tuning experiments (See Table 7) is 0.99 (p=2.0e-16).

Table 12: Reranker experiment results

| Model | All (%) | | | | Heldout Test (%) | | | |
|---|---|---|---|---|---|---|---|---|
| | Pr | Rc | F1 | Acc | Pr | Rc | F1 | Acc |
| majority baseline | 66.8 | 1.00 | 80.1 | 66.8 | 67.1 | 1.00 | 80.3 | 67.1 |
| bge-reranker-base | 77.4 | 59.8 | 67.5 | 62.1 | 76.5 | 58.4 | 66.2 | 62.4 |
| bge-reranker-large | 77.5 | 56.3 | 65.2 | 60.5 | 78.1 | 56.2 | 65.4 | 62.4 |
| bge-reranker-v2-m3 | 75.7 | 73.6 | 74.6 | 67.1 | 76.2 | 71.9 | 74.0 | 68.1 |
| jina-reranker-v2-base-multilingual | 76.4 | 65.1 | 70.3 | 63.8 | 81.1 | 67.4 | 73.6 | 69.5 |
| Zeroshot Llama 3.1 | 77.9 | 81.7 | 79.7 | 72.7 | 80.5 | 78.7 | 79.5 | 74.5 |
| Zeroshot Qwen3-Reranker-8B | 73.7 | 98.7 | 84.4 | 76.0 | 71.9 | 97.8 | 82.9 | 74.5 |
| Finetuned-qwen3-reranker | n/a | n/a | n/a | n/a | 87.0 | 89.9 | 88.4 | 85.1 |

Evaluation metrics (%) for various reranker models. For zero-shot rerankers, we show performance on all datapoints and a heldout test set, for fine-tuned models, we only show performance for 20% randomly held-out datapoints.

Table 13: Examples queries from different datasets

| Dataset | Queries |
|---|---|
| Barexam QA | Paul, the Plaintiff in a personal injury action, called Wes as a witness to testify that Dan's car, in which Paul had been riding. ran a red light. Wes, however, testified that Dan's car did not run the light. Paul then called Vic to testify that Dan's car did run the light. The trial judge should rule that Vic's testimony is |
| | Paul, the Plaintiff in a personal injury action, called Wes as a witness to testify that Dan's car, in which Paul had been riding. ran a red light. Wes, however, testified that Dan's car did not run the light. On cross-examination of Vic, Dan's attorney asked if Vic was drunk at the time he witnessed the accident. and Vic responded, "No I have never in my life been drunk." Dan's attorney then sought to prove by Yank that Vic was drunk on New Year's Eve two years before the accident. The trialjudge should rule that Yank's testimony is |
| | Paul, the Plaintiff in a personal injury action, called Wes as a witness to testify that Dan's car, in which Paul had been riding. ran a red light. Wes, however, testified that Dan's car did not run the light. Dan called Zemo as a witness and asked him if he knew Vic's reputation for veracity in the community where Vic resided. The trialjudge should rule that this question is |
| LePaRD | Hill, 131 F.3d at 1062 (quoting Mathis, 963 F.2d at 408). In this case, because Smith's and Cook's escape indictments are devoid of detail, and Thomas' indictments were never proffered to the court, the parties agree that we should look no further than the statutory language. See Taylor, 495 U.S. at 600, 110 S.Ct. at 2159; United States v. Luster, 305 F.3d 199, 202 (3d Cir.2002); United States v. Pierce, 278 F.3d 282, 287 (4th Cir.2002). That is, the offenses defined by those statutes do not have " |
| | Aldy ex rel. Aldy v. Valmet Paper Mach., 74 F.3d 72, 75 (5th Cir. 1996) (emphasis omitted) (quoting Stena Rederi, 923 F.2d at 386). The third element ensures there is a jurisdictional nexus with the United States. Arriba, 962 F.2d at 533. The "direct effect" requirement involves a determination of whether the acts the suits are "based upon" had " |
| | The final element of the "commercial activity" exception poses the issue remaining on this remand, namely, in light of Weltover, did the detention of the aircraft cause " |
| Synthetic Naive | What happens when a country refuses to honor an international arbitration award? |
| | Do plaintiffs in *anonymized cases* involving defined contribution plans have standing to sue for breach of fiduciary duty claims under Section *anonymized*? |
| | How do age-related developmental differences impact the evaluation of children in social interactions and behaviors? |
| Synthetic Optimized | waiver of rights by defendant with limited English proficiency |
| | what is the current status of the law on large capacity magazines? |
| | what are the circumstances under which a remand is appropriate? |
| Real-world queries | anonymized statute |
| | what is the definition of probable cause? |
| | Is anonymized case still good law? |

