# OpenReview forum: "Legal Retrieval for Public Defenders"
_TMLR — Under review for TMLR_

### Review · Reviewer_WHV7 · 2026-03-16

**Summary Of Contributions:**

The paper introduces a realistic legal retrieval benchmark for public defense, motivated by collaboration with a public defender office and centered on retrieving relevant passages from prior appellate briefs rather than solving more artificial legal QA tasks. It releases a manually annotated dataset of real defender queries and relevant paragraphs, together with a taxonomy of defender search behaviors. Beyond dataset construction, the paper shows that existing legal retrieval benchmarks transfer poorly to this setting, while domain-specific techniques such as legal query expansion, curated synthetic data, and legal domain adaptation can substantially improve retrieval quality. Overall, the work makes a useful contribution by grounding legal NLP evaluation in an authentic, high-stakes workflow and highlighting the importance of domain mismatch in legal retrieval.

**Audience:**

Yes

**Audience Explanation:**

Yes. The paper should be of interest to at least part of the TMLR audience, especially researchers working on information retrieval, legal NLP, and human-centered AI.

**Broader Impact Concerns:**

A Broader Impact Statement should be added to the paper.

Because the system targets public defense, errors or overreliance could affect high-stakes legal work, and the paper would benefit from a short, explicit discussion of deployment risks, privacy/confidentiality safeguards, and the need for human verification. The authors partly mitigate this by using anonymized documents for the released dataset and excluding queries with personally identifiable information.

**Claims And Evidence:**

Yes

**Claims Explanation:**

Mostly yes. The main empirical claims are generally supported by the experiments: the paper introduces a realistic public-defense retrieval benchmark, evaluates several retrieval models, and provides consistent evidence that existing legal retrieval benchmarks transfer poorly to this setting, while domain-specific interventions such as curated synthetic data, query expansion, and legal adaptation improve performance. These claims are backed by quantitative retrieval results and qualitative analysis of query types and workflow needs.

That said, the evidence is more convincing for the benchmark and domain-mismatch claims than for broader conclusions about practical deployment or general legal retrieval. The dataset is still relatively small, and some of the stronger claims about real-world utility would benefit from more extensive user studies.

**Requested Changes:**

1. Clarify resource release and reproducibility (This is critical). The paper repeatedly states that the dataset, fine-tuned models, and query taxonomy will be released, but the submission does not clearly specify a public link, release plan, or availability details. The authors should make the release status explicit and provide enough information for reproducibility.

2. Strengthen the description of data collection and annotation quality. Since the main contribution is a benchmark, the paper should more clearly document dataset construction, annotation protocol, inter-annotator agreement, or quality-control procedures, and any filtering or privacy constraints that shaped the final dataset.

3 Calibrate the scope of the claims. The central evidence supports the usefulness of the benchmark and the domain-mismatch finding, but broader claims about general legal retrieval or practical deployment should be stated more carefully unless supported by additional evidence.

---

> ### Author Response · Authors · 2026-05-08
> **Review Response for Reviewer WHV7**
>
> We thank reviewer WHV7 for the overall positive assessment of the work (“the work makes a useful contribution by grounding legal NLP evaluation in an authentic, high-stakes workflow”), but also the suggested changes which helped us think through dataset release and address important concerns around dataset release.
>
> **Dataset resource release**
>
> - We host an anonymized version (anonymized with gemma4-31B) of the dataset and minimal replication code in this anonymized github repo: https://anonymous.4open.science/r/anonymized-public-defender-retrieval-782E/README.md Upon publication, we will release the full dataset with anonymized party names (and other PII  related to parties anonymized).
>
> - We changed the line “we release a manually annotated dataset, fine-tuned models, and a taxonomy of defender search queries.” to “we release a manually annotated dataset, fine-tuned models and replication code.”, as we summarize all the relevant information of the query taxonomy in Section 4. We will host fine-tuned models on hugging face and replication code on github, with links to both in the paper.
>
> **Anonymization**
>
> - As part of the revisions, we anonymized party-related PII using a Gemma-4 model. We describe the exact process and manual evaluation in Section 3.2. While this process is not perfect, it errs on the side of overanonymization, and does not affect retrieval performance (Spearman R between zero-shot correlations in the anonymized and non-anonymized are 1.0 (perfect correlation); for fine-tuning experiments, the correlation is 0.99. We reran all experiments, and updated all results and correlations in Section 5 and the Appendix in the revised paper. We also included a new appendix Table where we present and compare anonymized to non-anonymized results.
>
> **Quality Control**
>
> - We updated Section 3.2 to include information about anonymization procedures. An independent annotator not connected to the annotation process re-annotated 100 query / paragraph pairs in the released dataset. Inter-annotator agreement between this new annotator (with law school experience) and the existing annotations results in a Cohen’s Kappa of 0.36, indicating fair agreement. Most of the disagreements (66\%) can be explained by the independent annotator being more lenient, and annotating slightly relevant paragraphs–compared to the author team which was more conservative and mostly annotated highly relevant paragraphs instead (which is more in line with the proprietary annotations by public defenders as part of the BriefBank evaluation round from which we have the queries).
>
> - Crucially, the direction of disagreement (being more lenient instead of rejecting annotations in the dataset) in our view does not undermine the benchmark's ability to rank retrieval systems. A more conservative annotation threshold simply results in a harder benchmark, but models that retrieve more relevant paragraphs will still score higher under either annotation standard. More importantly though, we believe the high correlations between the proprietary set annotated by public defenders and our released version are the strongest signal for quality control, indicating validity of the benchmark for assessing methods on public defense retrieval. We added the IAA validation to Section 3.2
>
> **Calibrate the scope of the claims**
>
> - We revisited all claims made about general legal retrieval and practical deployment, and replaced them with more hedged versions. These include:
>
> - Abstract: “Our work [...] provides a starting point for leveraging AI” → “Our work [...] illustrates one approach to applying AI”
>
> - Page 2: “the BriefBank helps to streamline brief drafting” → “the BriefBank is designed to streamline brief drafting”
>
> - Page 3: “Combined, these artifacts can help” → “Combined, these artifacts could help” and "allow situating NLP research in impactful, real-world applications" → "allow situating NLP research in real-world applications."
>
> - Section 3.2: "the PD dataset captures relevant signals about public defender search and real-world legal retrieval" --> "the PD dataset captures relevant signals about public defender search"
>
> - Section 4: “These can point to common failure cases in realistic legal retrieval.” → “These can point to common failure cases, and inform future research in public defense retrieval.”
>
> - Section 6.1 Rewritten to “Next, we construct a taxonomy about search objectives of defenders and what search strategies can be employed to answer these queries. Similar to datasets like WildChat Zhao et al. (2024), which make real-world chatGPT conversations accessible for research, we hope our queries and taxonomy can inform future work in more realistic legal retrieval settings.”
>
> - We have rewritten Section 6.2
>
> - Conclusion: "legal retrieval for public defense is constrained." → "may be constrained."
>
> **Broader Impact**
> - We added a discussion about broader impact after the discussion.

---

> > ### Comment · Reviewer_WHV7 · 2026-07-20
> >
> > Dear authors,
> >
> > Thanks for your response.
> >
> > The anonymized GitHub repository currently appears to contain only a README file, with no dataset or replication code available. Could you please update the repository accordingly and ensure that the anonymized dataset and minimal replication code described in the response are accessible for review?

---

> > > ### Author Response · Authors · 2026-07-20
> > > **New anonymized GitHub**
> > >
> > > Sorry about this, it seems like the previous version automatically anonymized an intermediary version of the repo. This here should be the up-to-date version.
> > >
> > > https://anonymous.4open.science/r/anonymized-public-defender-retrieval-EB0C/
> > >
> > > (The corpus is uploaded as two zip files, anonymous GitHub afaik has a 50M max file size). Happy to address other questions / concerns too!

---

> > > > ### Comment · Reviewer_WHV7 · 2026-07-20
> > > >
> > > > Thanks for providing the updated repository!

---

### Review · Reviewer_4Ywg · 2026-03-19

**Summary Of Contributions:**

The paper introduces BriefBank, a retrieval system designed for an OPD. This framework allows defenders to search across an internal corpus of past briefs to quickly find relevant legal arguments and citations.

---

### Strengths:
- The paper is specially novel and can be used in an actual system used by public defenders, with authentic queries and annotations. Moreover, this work shows that existing legal benchmarks and synthetic data does not perform well on public defense retrieval, and provides a useful taxonomy.
- As such, the paper is well-written and easy to follow.

### Weaknesses:
- The new PD dataset is built on only ~170 queries. While high-quality and human-annotated, this may quite small for robustly evaluating and fine-tuning modern embedding models.
- In the legal domain, as stated in Section 2.1, hallucinations are catastrophic. There is a lack of evaluation of the LLM generation component throughout the paper.  They are missing experiments evaluating the factual accuracy, hallucination rate, and overall utility of these LLM-generated summaries.
- The authors also reports that this combined pipeline improves performance - could the authors provide an ablation study for the same?
- OPD spans 25 years, while PD is 2023 to 2025. Could the authors evaluate retrieval performance separately on older vs newer OPD documents to verify generalization?

---

**Audience:**

Yes

**Audience Explanation:**

Although I do not have expertise in this specific field of legal retrieval, this work presents a highly compelling use case. I am confident that the broader TMLR audience will find this work both interesting and valuable.

**Broader Impact Concerns:**

The paper currently lacks a Broad Impact statement. Since this research involves deploying LLMs and retrieval systems in public defense, a discussion of ethical implications is required.

**Claims And Evidence:**

Yes

**Claims Explanation:**

Majority of the claims in the paper have been experimentally verified. However, some ablation studies and further experiments would definitely strengthen the paper.

**Requested Changes:**

Major changes have been listed in the Weaknesses. Some minor comments are listed below:
- A few citations are misformatted, please use /citep wherever needed.

---

> ### Author Response · Authors · 2026-05-08
> **Review Response for Reviewer 4Ywg**
>
> We thank the reviewer 4Ywg for the positive assessment of the paper (“paper is specially novel”, “paper is well-written and easy to follow” and “this work presents a highly compelling use case”, and also the provided feedback and suggestions which helped us to create a more compelling, revised paper. In the following, we respond to the reviewer comments in more detail:
>
> - The paper currently lacks a Broad Impact statement.
>
> We added a broader impact statement after the conclusion, discussing the intended use case, considerations in the BriefBank, and limitations of the released dataset.
>
> - The new PD dataset is built on only ~170 queries. While high-quality and human-annotated, this may quite small for robustly evaluating and fine-tuning modern embedding models.
>
> We changed the abstract “we release a new, realistic dataset” → “we release a new, realistic evaluation dataset” and the contribution bullet point “We construct an accompanying dataset” → “We construct an accompanying evaluation dataset”. We believe it’s already clear in the current version that we do not fine-tune embedding models on these queries, but use synthetic data throughout for all fine-tuning experiments. We are aware of the rather small dataset size, and hence already included various robustness checks (error bars, correlations between proprietary dataset and released dataset, evaluation metrics) to ensure that the evaluation set is as robust as possible, although the N is small.
>
> - hallucinations are catastrophic. There is a lack of evaluation of the LLM generation component throughout the paper.
>
> We added one paragraph in the broader impact statement justifying why we use LLM-generated summaries in the BriefBank, and discuss safeguards we implemented to ensure that these are not treated as reliable information, but only to quickly decide whether a returned document might be relevant.
>
> - The authors also reports that this combined pipeline improves performance - could the authors provide an ablation study for the same?
>
> We believe the core of our work to be a public defender retrieval task, and more broadly a more realistic legal retrieval benchmark. We already report individual aspects of the pipeline (overall retrieval performance, results of query expansion, reranker performance) in Section 5 and have rewritten the paragraph in page 2 in the intro (starting with “On the other hand, using larger embedding models [...]” → starting now with “On the other hand, we improve recall by (1) using more recent and larger embedding models” ) to not imply combined pipeline performance improvement without supporting ablation results.
>
> We do hint at other pipeline components in the BriefBank (e.g., AI-generated summaries), but we believe these are less relevant for the retrieval-oriented paper and justified their use in the Broader Impact Section. A full end-to-end ablation of all these components likely requires controlled user studies—which we think is beyond the scope of the current retrieval-oriented paper. We plan to address this in future work examining the broader impact of AI on public defense practice.
>
> - OPD spans 25 years, while PD is 2023 to 2025. Could the authors evaluate retrieval performance separately on older vs newer OPD documents to verify generalization?
>
> We implemented these experiments, and added a paragraph in Section 5 (in Robustness of the PD Dataset): “Correlations between the released PD dataset and the OPD dataset are not driven by old OPD retrieval targets, but also generalize to more recent retrieval targets.” We also added Appendix table 6 reporting results on older vs newer OPD documents.
>
> - A few citations are misformatted
>
> We did another pass and hope that we have fixed all misformatted citations.

---

### Review · Reviewer_WdER · 2026-06-22

**Summary Of Contributions:**

The author studies legal retrieval for public defenders and introduces BriefBank, a retrieval-based system that returns relevant passages from past briefs and legal documents instead of directly generating legal answers. They also build a public PD benchmark with real defender queries and annotated relevant paragraphs. A key finding is that existing legal retrieval benchmarks do not transfer well to this setting, while domain-specific synthetic data, query expansion, and fine-tuned rerankers improve performance. I think the paper addresses an important real-world problem, but its main contribution seems to be the dataset and system construction rather than a new retrieval method. From a methodological perspective, the novelty is somewhat limited.

**Audience:**

Yes

**Audience Explanation:**

Yes. I think some TMLR readers would be interested in the paper because it studies a realistic legal retrieval setting and shows that existing legal retrieval benchmarks may not transfer well to real public-defense queries.

**Broader Impact Concerns:**

I do not see major broader impact concerns beyond those already discussed in the paper.

**Claims And Evidence:**

Yes

**Claims Explanation:**

The paper studies an interesting and important real-world legal retrieval problem with useful empirical findings, but my main concern is that the contribution is mostly in dataset and system construction, while the methodological novelty is relatively limited. I've listed my thoughts below:

Strength:

1.	The paper studies an important and realistic problem. Public defense is a high-stakes setting, and the paper gives a convincing motivation for why retrieval is safer and more useful than directly generating legal answers.

2.	The collaboration with a real public defender office is a strong point. Since the queries come from actual defenders, the benchmark feels more realistic than many existing legal retrieval datasets.

3.	I like the empirical finding that existing legal retrieval benchmarks do not transfer well to this setting. This is useful and interesting, and it clearly shows the gap between academic legal benchmarks and real-world legal search needs.

Weakness/questions:

1.	For an academic journal, the method contribution is somewhat limited and less novel. Most of the approach is a system and data pipeline combining existing retrievers, rerankers, synthetic data generation, and query expansion, rather than a new retrieval algorithm.

2.	I also have some concerns about the model-assisted annotation pipeline. The candidate paragraphs are first filtered by retrieval models, GPT-4, and a fine-tuned reranker before human annotation. This makes the dataset construction efficient, but it may also introduce selection bias: relevant paragraphs that are filtered out early would never be considered by human annotators.

3.	The generalization is still unclear to me. The benchmark is based on one public defender office and one state/jurisdiction, so it is not obvious whether the findings will transfer to other defender offices or other legal domains.

4.	Some claims about real-world utility feel a bit stronger than the evidence. The paper shows improvements in retrieval metrics, but it does not fully evaluate whether the system actually improves downstream performance such as defenders’ workflow, writing quality, or practical decision-making.

**Requested Changes:**

Please see weakness above.

---

> ### Author Response · Authors · 2026-06-26
> **Review Response for Reviewer WdER**
>
> We thank reviewer WdER for the overall positive review (“paper studies an important and realistic problem”, “the paper gives a convincing motivation for why retrieval is safer and more useful”, “collaboration with a real public defender office is a strong point”, “I like the empirical finding that existing legal retrieval benchmarks do not transfer well to this setting”). We also appreciate the raised weaknesses and questions. We submitted a revised version and believe we addressed these. We will engage with the requested changes one by one in the following:
>
> - For an academic journal, the method contribution is somewhat limited and less novel. Most of the approach is a system and data pipeline combining existing retrievers, rerankers, synthetic data generation, and query expansion, rather than a new retrieval algorithm.
>
> We agree with this comment to some extent. While we do not come up with new retrieval algorithms, we believe we examined all the most common retrieval methods, including zero-shot, transfer learning (training on BarExam and LePaRD), synthetic data, curated synthetic data, query expansion, and domain adaptation through pre-training ModernBERT on court opinions.
>
> In line with the review, we believe the main contribution is the identified use case, its motivation and the resulting benchmark (which were all positively assessed by the reviewer too). Most reported results seem to be in line with expectations, which in turn validates the data contribution. We believe it's reasonable to leave it to future work, by the authors or other researchers, to use the PD dataset to develop and validate new retrieval algorithms.
>
> - The generalization is still unclear to me. The benchmark is based on one public defender office and one state/jurisdiction, so it is not obvious whether the findings will transfer to other defender offices or other legal domains.
>
> In the revised version, we have included “Section 8. Broader Impact Statement” where we acknowledge exactly this point: “However, we acknowledge that the benchmark stems from a single office in one U.S. state. Generalization to other jurisdictions or office contexts should thus be verified. Moreover, the benchmark might be biased (1) towards retrieval targets specific to anonymized state, (2) specific annotation idiosyncrasies of the annotating defenders and the author team, and (3) we relied on automatic filtering using an ensemble of GPT-4o and a fine-tuned Qwen3-8B reranker to further reduce the number of  paragraphs we manually reviewed. These limitations need to be considered in future usage.”
>
> - I also have some concerns about the model-assisted annotation pipeline. The candidate paragraphs are first filtered by retrieval models, GPT-4, and a fine-tuned reranker before human annotation. This makes the dataset construction efficient, but it may also introduce selection bias: relevant paragraphs that are filtered out early would never be considered by human annotators.
>
> While we also agree with this comment to some extent, we note that this critique is almost universally true for every retrieval benchmark. The author team conducted many spot checks to ensure that there are only a few randomly valid search results (1) not surfaced by initial search candidates, and (2) not discarded by our filtering strategy using GPT-4 and the fine-tuned reranker as a judge. Second, we show very high correlation in model and method rankings with the proprietary dataset annotated by public defenders. Search candidates in this dataset did not go through the same LLM pre-filtering stage.
>
> We also address this in the revised version in “Section 8. Broader Impact Statement”.
>
> - Some claims about real-world utility feel a bit stronger than the evidence.
>
> Other reviewers have raised similar concerns. In response, We revisited all these claims, and replaced them with more hedged versions. For example: Abstract: “Our work [...] provides a starting point for leveraging AI” → “Our work [...] illustrates one approach to applying AI”; Page 2: “the BriefBank helps to streamline brief drafting” → “the BriefBank is designed to streamline brief drafting.” We point to our response to reviewer WHV7 for the full list of revised claims.